# Extracellular Vesicles Derived from Bone Marrow in an Early Stage of Ionizing Radiation Damage Are Able to Induce Bystander Responses in the Bone Marrow

**DOI:** 10.3390/cells11010155

**Published:** 2022-01-04

**Authors:** Dávid Kis, Ilona Barbara Csordás, Eszter Persa, Bálint Jezsó, Rita Hargitai, Tünde Szatmári, Nikolett Sándor, Enikő Kis, Katalin Balázs, Géza Sáfrány, Katalin Lumniczky

**Affiliations:** 1National Public Health Center, Department of Radiobiology and Radiohygiene, Unit of Radiation Medicine, 1097 Budapest, Hungary; kis.david@osski.hu (D.K.); csordas.ilona@osski.hu (I.B.C.); persa.eszter@osski.hu (E.P.); hargitai.rita@osski.hu (R.H.); szatmari.tunde@osski.hu (T.S.); sandor.nikolett@osski.hu (N.S.); kis.eniko@osski.hu (E.K.); balazs.katalin@osski.hu (K.B.); safrany.geza@osski.hu (G.S.); 2Doctoral School of Pathological Sciences, Semmelweis University, 1085 Budapest, Hungary; 3Doctoral School of Biology and Institute of Biology, Eötvös Loránd University, 1053 Budapest, Hungary; jezso.balint@ttk.elte.hu; 4Research Centre for Natural Sciences, Institute of Enzymology, 1117 Budapest, Hungary

**Keywords:** extracellular vesicles, ionizing radiation, bystander effects, bone marrow, stem cells, miRNA

## Abstract

Ionizing radiation (IR)-induced bystander effects contribute to biological responses to radiation, and extracellular vesicles (EVs) play important roles in mediating these effects. In this study we investigated the role of bone marrow (BM)-derived EVs in the bystander transfer of radiation damage. Mice were irradiated with 0.1Gy, 0.25Gy and 2Gy, EVs were extracted from the BM supernatant 24 h or 3 months after irradiation and injected into bystander mice. Acute effects on directly irradiated or EV-treated mice were investigated after 4 and 24 h, while late effects were investigated 3 months after treatment. The acute effects of EVs on the hematopoietic stem and progenitor cell pools were similar to direct irradiation effects and persisted for up to 3 months, with the hematopoietic stem cells showing the strongest bystander responses. EVs isolated 3 months after irradiation elicited no bystander responses. The level of seven microRNAs (miR-33a-3p, miR-140-3p, miR-152-3p, miR-199a-5p, miR-200c-5p, miR-375-3p and miR-669o-5p) was altered in the EVs isolated 24 hour but not 3 months after irradiation. They regulated pathways highly relevant for the cellular response to IR, indicating their role in EV-mediated bystander responses. In conclusion, we showed that only EVs from an early stage of radiation damage could transmit IR-induced bystander effects.

## 1. Introduction

The effects of ionizing radiation (IR) on the living cells can be seen not only on cells directly hit by the radiation beam, but also on the neighboring and sometimes distant cells in the same tissue [1,2,3]. This phenomenon is called the radiation-induced bystander effect. Signals transmitted by damaged cells via gap junctions or factors released into the extracellular space, such as reactive oxygen species, chemokines, cytokines, free microRNAs and other danger signals, are important elements in the mechanism of radiation-induced bystander effects [4,5]. In recent years, it has been proposed that extracellular vesicles (EVs) can also contribute to the transmission of bystander signals [6]. EVs are small (50–1000 nm), membrane-coated bodies, actively released by most cell types, playing a role in intercellular communication [7,8]. Their cargo consists of various nucleic acids, mainly RNA, microRNA, proteins, lipids and small metabolites originating from the parental cell [9,10,11,12,13], and by travelling in the extracellular space and blood, they are able to transfer signals to other, distant cells [14,15], facilitating intercellular communication. It has been also shown that under stress conditions, the EV release of the cells increases [16,17].

The role of EVs in radiation-induced bystander effects has been studied in the last decade, although most of these studies were done in vitro. It has been shown that EVs from irradiated cells were capable of promoting radiation-related effects in EV-recipient cells. EV-mediated bystander effects were demonstrated after both low-dose and high-dose irradiation [18,19,20]. The role of EVs in different diseases, such as cancer, cardiovascular diseases or infectious diseases, was also investigated by several research groups, but in the majority of these studies, blood or urine was used for EV isolation. Therefore, established protocols exist for the isolation of EVs from these fluids [21,22,23,24]. Although, technically, any kind of extracellular body fluid is suitable for EV isolation, other tissues were less commonly used. Recently, we isolated EVs from bone marrow (BM) and have demonstrated that BM-derived EVs mediate radiation-induced bystander effects in the spleen, BM and plasma [25,26,27]. In these studies, we reported the establishment and validation of an in vivo bystander model using C57BL/6 mice, suitable to investigate the role of BM-derived EVs in mediating systemic bystander effects. In the present paper we investigated acute and long-lasting radiation-induced bystander effects mediated by EVs in different hematopoietic stem and progenitor cell subpopulations, and analyzed how long after irradiation EVs maintain their capacity to transfer bystander effects.

## 2. Materials and Methods

### 2.1. Murine Model and Irradiation Protocol

Nine- to twelve-week-old male C57/BL6 mice were used in all experiments. Mice were kept and treated in accordance with the respective Hungarian law for animal welfare and the European 2010/63/EU directives and regulations. All animal studies were approved and permission was issued by Budapest and Pest County Administration Office Food Chain Safety and Animal Health Board (ethical permission number: PE/EA/392-7/2017).

The mice were total-body irradiated with 0 (control), 0.1, 0.25 and 2Gy X-rays using X-RAD 225/XLi X-ray source (Precision X-Ray, Madison, CT, USA). For each dose, 12–20 mice were used. Experimental groups contained randomly chosen mice aged between 9 and 12 weeks.

### 2.2. Isolation of Mouse Bone Marrow Cells and Splenocytes

BM was isolated from the tibia and femur of mice as described previously [27]. Briefly, BM tissue was flushed out from the diaphysis and suspended in phosphate-buffered saline (PBS). BM cell (BMC) suspensions were made by thorough pipetting. Cells were pelleted by centrifugation at 400 g, 4 °C for 10 min and passed through a 40 μm cell strainer. Cell pellets were used for phenotype analysis while supernatants were used for EV isolation.

Spleens were processed as described previously [27]. Briefly, spleens were mechanically disaggregated and single-cell suspensions were pelleted in PBS. Red blood cells were removed by incubation of the pellets in 5 mL lysis buffer containing 155 mM NH4Cl, 10 mM KHCO3, and 0.1 mM EDTA for 5 min. Cells were washed with PBS and passed through a 40 μm cell strainer.

The BM and spleen of irradiated and bystander mice were processed individually. The number of viable BMCs and splenocytes was determined by trypan blue exclusion. Cells were used for immune phenotyping of different subpopulations, apoptosis and migration analysis immediately after isolation.

For analysis of BMC migration into the spleen, a separate group of mice was injected intraperitoneally (i.p.) with 166 μg of AMD3100/Plerixafor (MCE MedChemExpress, Monmouth Junction, NJ, USA) dissolved in sterile PBS and humanely killed 2.5 h after the injection. AMD3100-treated mice served as positive controls, since AMD3100 is a selective CXCR4 antagonist and hematopoietic stem cell mobilizer in the clinic [28].

### 2.3. Isolation, Validation and In Vivo Transfer of EVs

EVs were prepared from the pooled BMC supernatants of at least three mice/treatment group. EVs were isolated 4 h, 24 h or 3 months after irradiation using the ExoQuick-TC kit (System Biosciences, Palo Alto CA, USA) following the manufacturer’s instructions as described previously [27]. Briefly, supernatants were incubated with ExoQuick-TC solution at 4 °C overnight followed by centrifugation at 1500 *g* for 30 min. The pelleted EVs were suspended in 140 μL of PBS. A GE Healthcare PD SpinTrap G-25 desalting column (GE Healthcare, Life Sciences, WI, USA) was used to remove ExoQuick polymers from the EV solutions.

The size and concentration of extracellular vesicles was measured using a tunable resistance pulse sensing (TRPS) system (EXOID from Izon Science™, Lyon, France) in which samples were driven through nanopores with two different pore sizes (NP150 and NP250) by applying a combination of three different pressures and constant voltage. The resistive pulse or blockade signal caused by each particle was detected and measured by the instrument. Blockade magnitude was directly proportional to the volume of each particle [29], while blockade frequency indicated particle concentration [30]. For calibration carboxylated polystyrene nanoparticles/beads (TKP200 from Izon Science™, Lyon, France) were used.

For transmission electron microscopy (TEM) analysis, EVs were processed as described in [31]; namely, samples were applied to formvar-coated nickel grids and stained with 2% uranyl acetate (*v*/*v*) solution for 5 min. Grids were air-dried and viewed using a JEOL TEM 1011 TEM (JEOL, Peabody, MA, USA) operated at 80 kV. The camera used for image acquisition was a Morada TEM 11 MPixel from Olympus (Olympus, Tokyo, Japan) using iTEM5.1 software for metadata analysis.

The protein content of EVs was measured with a Bradford protein assay kit (Thermo Fisher Scientific, Waltham, MA, USA) using a Synergy HT (Agilent, Santa Clara, CA, USA) plate reader.

EV-specific protein markers were investigated by Western blot analysis as described in [27], where 40 μg of EVs were lysed with radioimmunoprecipitation assay (RIPA) lysis buffer containing 2% protease inhibitors (Sigma-Aldrich, Darmstadt, Germany) and equal amounts of protein lysates were loaded and electrophoresed on 10% sodium dodecyl sulfate-polyacrylamide (SDS-PAGE) gel and transferred to polyvinylidene fluoride (PVDF) membranes (Bio-Rad, Hercules, CA, USA). As controls, murine BM whole cell lysates were used, which were treated in the same way as EV protein lysates. The Prism Ultra Protein Ladder (Abcam, Cambridge, UK) was used as a protein standard. The following antibodies were used: anti-mouse CD9, TSG101, annexin V and calnexin, all purchased from Abcam. Protein lysates were incubated with the antibodies at room temperature for 1.5 h, followed by incubation with goat anti-rabbit secondary antibody conjugated to horseradish peroxidase (Abcam) for 1 h. Membranes were washed in Tris-buffered saline-tween buffer three times, and protein bands were visualized using 3,3′-diaminobenzidine substrate (Sigma-Aldrich), by the chromogenic method.

Ten μg of EVs from directly irradiated mice suspended in 140 μL PBS were injected into the tail vein of unirradiated mice. Mice injected with EVs and receiving no direct irradiation represent the bystander group. Both directly irradiated and EV-recipient mice were humanely killed 4 h, 24 h or 3 months after treatment.

### 2.4. Immune Phenotypical Analysis of BMCs and Splenocytes

The following directly labelled anti-mouse monoclonal antibodies were used: CD90.2-PE and CD45-PE/Cy7 for lymphoid progenitors (LP), Gr1-AF647 and CD11b-PE for granulocytes/monocytes (GM), Lineage Cocktail (CD3, Gr1, CD11b, CD45R, Ter119)-PB, Sca1-PE, cKit (CD117)-BV785, CD34-AF647, CD135-PE/Cy5 for hematopoietic stem and progenitor cells (HSPCs) and Lineage Cocktail-PB, Sca1-PE and CD44-AF647 for mesenchymal stem/stromal cells (MSCs), all purchased from SONY (SONY Biotechnology, San Jose, CA, USA). Individual cell populations were identified based on the following phenotypes: HSPCs: Lin-Sca1+cKit+ cells, long-term HSCs (LT-HSCs) Lin-Sca1+cKit+CD34-CD135- cells, short-term HSCs (ST-HSCs): Lin-Sca1+cKit+CD34+CD135- cells, multipotent progenitors (MPPs): Lin-Sca1+cKit+CD34+CD135+ cells, LPs: CD45+CD90.2+ cells, GMs: Gr1+CD11b+ cells, and MSCs: Lin-Sca1+CD44+ cells. A detailed gating strategy for the identification of the above cell populations is shown in Figure 1.

Single-cell suspensions were incubated with the fluorescently labelled antibodies in PBS containing 1% bovine serum albumin (BSA) at 4 °C for 30 min. Measurements were performed by a CytoFLEX (Beckman Coulter, Brea, CA, USA) flow cytometer. Data acquisition and analysis was performed using the CytExpert software version 2.3.0.84 (Beckman Coulter, Brea, CA, USA).

### 2.5. Analysis of Apoptosis

Apoptosis was detected by the TUNEL assay using the Mebstain Apoptosis Kit Direct (MBL, Nagoya, Japan) in the freshly isolated and phenotypically labelled BMCs as described above. Briefly, cells were permeabilized with 1 mL ice-cold 75% ethanol at 4 °C for 20 min, followed by fixation in 1 mL 1% PFA at 4 °C overnight and then pelleted. Cells were incubated with 27 μL of terminal deoxy-nucleotidyl transferase (TdT) buffer, 1.5μL of FITC-dUTP and 1.5 μL TdT enzyme per sample at 37 °C for 1 h. Before analysis of HSPCs and MSCs, the lineage-positive BMCs were removed by magnetic sorting with a Direct Lineage Cell Depletion Kit, mouse (Miltenyi Biotec, Auburn, CA, USA) following the manufacturer’s instructions. The purity of the lineage-negative cells after magnetic separation was above 92% in every sample.

The proportion of apoptotic cells was determined by flow cytometry using a FACSCalibur flow cytometer (Becton Dickinson, Franklin Lakes, NJ, USA). The Cell-Quest™ Pro data acquisition and analysis software version 4.0.2 was used for data analysis (Becton Dickinson).

### 2.6. Analysis of miRNA Expression from BM-Derived EVs

EVs were isolated as described above. The following miRNAs were examined: mmu-mir-152-3p, mmu-mir-199a-5p, mmu-mir-375-3p, mmu-miR-33-3p, mmu-miR-140-3p, mmu-miR-200c-5p, mmu-mir-744-3p, mmu-mir-669o-5p.

Total RNA was extracted from EVs by using the Qiagen RNeasy Plus Mini Kit (Qiagen, Hilden, Germany) according to the manufacturer’s instructions. Briefly, EVs were lysed in Buffer RLT Plus, then the homogenized lysate was centrifuged using a Qiagen genomic DNA eliminator spin column (≥8000 g for 30 s). The flow-through was treated with 1.5 volumes of 100% ethanol and centrifuged using a Qiagen RNeasy mini spin column (≥8000 g for 30 s), the column was washed twice with 500 µL buffer RPE and centrifuged ≥ 8000 g for 15 s, dried for 1 min and the total RNA (containing miRNA) was eluted with 50µL of RNase-free water.

Complementary DNA (cDNA) synthesis and miRNA expression analysis were carried out by using a miRCURY™ LNA™ miRNA PCR System (Qiagen). Firstly, 2 µL RNA with a concentration of 40 ng/µL was reverse-transcribed in 10 μL reaction volume. The cDNA was diluted 20-fold and assayed in 10 μL PCR reaction volume. The amplification was performed in a Rotor-Gene Q real-time PCR cycler (Qiagen). To determine the relative concentration of the screened miRNAs, the following PCR primers were used: hsa-miR-423-3p miRCURY LNA miRNA PCR Assay, hsa-miR-152-3p miRCURY LNA miRNA PCR Assay, hsa-miR-199a-5p miRCURY LNA miRNA PCR Assay, hsa-miR-375 miRCURY LNA miRNA PCR Assay, hsa-miR-33a-3p miRCURY LNA miRNA PCR Assay, hsa-miR-140-3p miRCURY LNA miRNA PCR Assay, hsa-miR-200c-5p miRCURY LNA miRNA PCR Assay, hsa-miR-744-3p miRCURY LNA miRNA PCR Assay, and mmu-miR-669o-5p miRCURY LNA miRNA PCR Assay, all purchased from Qiagen.

The amplification curves were analyzed by Rotor-Gene Q Series software (software version 2.1.0.9) both for determination of quantification cycles (Cq) and for melting curve (T_m_) analysis. In order for data to be included in further analysis, they had to meet the following criteria: appropriate melting curves, T_m_ had to be within known specifications for the assay, and the Cq value had to be ≤35.

The relative concentration of each miRNA was calculated by the Rotor-Gene Q software, where 0Gy irradiated samples were used as controls and mmu-miR-423-3p was used as a normalizer, since mmu-miR-423-3p was present in a constant and well detectable concentration in BM-derived EVs, which did not change after irradiation.

MiRNA pathway analysis was performed with DIANA-miRPath v.3.0 software, with the DIANA-microT algorithm. The pathway analysis was performed in genes union set, with a *p*-value of 0.05 and a MicroT threshold of 0.8; false discovery rate correction was also applied. To predict the potential target pathways, we used data provided by Kyoto Encyclopedia of Genes and Genomes (KEGG) [32].

### 2.7. Statistical Analysis

Statistical analysis was performed using the GraphPad Prism version 6.00 for Windows (GraphPad Software, La Jolla, CA, USA). The data are presented as mean ± standard deviation (SD). Since the sample size was low (below 50) the Shapiro-Wilk test was used for normality testing of the data series as recommended by Mishra et al. [33]. Statistical significance was determined using Student’s *t*-test. Data were considered statistically significant if the *p*-value was lower than 0.05.

## 3. Results

### 3.1. Validation of Bone Marrow-Derived EVs

BM-derived EVs were isolated from the BM supernatant of total-body irradiated mice as described previously [27]. EVs were validated by electron microscopy (Figure 2A), TRPS size and concentration measurement (Figure 2B) and the presence of EV-specific proteins (TSG101, annexin and CD9) (Figure 2C). Based on TRPS measurements, the average size of EVs was 150 nm and IR did not influence EV size distribution (Figure 2B/1). These data were supported by TEM measurements as well, showing classical EV morphology and vesicular structures in the anticipated size range (Figure 2A). Since irradiation strongly influenced BMC numbers, EV concentration was adjusted to the number of BMCs in each sample and particle numbers released by 10^6^ cells were calculated. The concentration of EVs isolated from the BM supernatant of mice 24 h after irradiation showed a dose-dependent increase, with a 2.85-fold (*p* = 0.056) and 4-fold (*p* = 0.026) increase in 0.25Gy and 2Gy EVs, respectively. Three months after irradiation, increased EV secretion in the BM of mice irradiated with 2Gy still persisted, but was milder (2.38-fold, *p* = 0.041) than EV secretion 24 h after irradiation (Figure 2B/2). EVs were characterized by Western blot analysis following minimal criteria suggested by Théry et al. [34]. EVs of both control and irradiated mice contained EV-specific proteins (TSG101, annexin V and CD9), and lacked calnexin, an endoplasmic reticulum marker (Figure 2C: lane 3-4-5).

### 3.2. Radiation-Induced Bystander Effects Are Elicited by BM-Derived EVs Isolated from Mice 24 Hours but Not 3 Months after Irradiation

EVs were isolated either from unirradiated or from total-body irradiated mice early (24 h) or late (3 months) after irradiation and were used to investigate bystander effects in unirradiated mice. The following treatment groups were designed for the study: (1) short-term bystander effects in mice treated with EVs isolated 24 h after irradiation, when bystander responses were followed 24 h after EV treatment; (2) long-term bystander effects in mice treated with EVs isolated 24 h after irradiation, when bystander responses were followed 3 months after EV treatment; (3) and bystander effects initiated 24 h after injection of EVs isolated 3 months after irradiation of mice (Figure 3). Alterations in the fraction of BM-stem and progenitor cell subpopulations and in mesenchymal stromal cells (MSCs) were studied in the three bystander treatment groups and changes were compared to the direct effects of IR. In order to simplify the description of the different treatment groups, we will use the following terms: 0Gy EVs indicating BM-derived EVs isolated from unirradiated mice, and 0.1Gy EVs, 0.25Gy EVs, 2Gy EVs indicating BM-derived EVs isolated from 0.1Gy, 0.25Gy and 2Gy irradiated mice, respectively.

#### 3.2.1. Alterations in the HSPC Pool

Relative HSPC numbers in directly irradiated mice compared to unirradiated controls decreased to 38%, 34% and 21% twenty-four hours after 0.1Gy, 0.25Gy and 2Gy irradiation, respectively (Figure 4A, grey bars). In mice receiving EVs isolated 24 h after irradiation, short-term bystander effects were similar to the effects of direct ionizing radiation, but changes were milder; relative HSPC numbers decreased to 65% and 60% after treatment with 0.25Gy EVs and 2Gy EVs compared to 0Gy EVs. (Figure 4A, red bars). If mice were treated with EVs isolated 3 months after BM irradiation, no statistically significant bystander response was measured in the HSPC population. However, it is important to note the increased inter-individual variability in the HSPC pool, which was not radiation-dependent (Figure 4A, yellow bars). Three months after irradiation, HSPC numbers in mice irradiated with low doses (0.1Gy and 0.25Gy) returned to the control values seen in unirradiated mice, while reduced HSPC numbers persisted in mice irradiated with 2Gy (Figure 4B, grey bars). Interestingly, long-term bystander effects induced by EVs isolated 24 h after irradiation manifested in strongly reduced HSPC numbers in mice treated with both low- (0.1Gy and 0.25Gy) and high (2Gy)-dose EVs (Figure 4B, blue bars). Changes were very similar to those seen in directly irradiated mice 24 h after irradiation.

Within the HSPC population, three major subpopulations can be identified: long-term HSCs (LT-HSCs), short-term HSCs (ST-HSCs) and multipotent progenitors (MPPs) [35]. Since these cells have distinct proliferative capacities, we were interested in investigating direct IR and EV-mediated bystander effects in the individual HSPC subpopulations as well. Phenotypical discrimination of these subpopulations was based on the presence or absence of CD34 and CD135 markers on Lin-Sca1+cKit+ HSPC as described in [36,37]. However, as has recently been shown, CD34+CD135- ST-HSCs contain various MPP subpopulations as well [38].

The tendency of changes in the pool of the individual HSPC subpopulations after irradiation and EV treatment was very similar to the unfractionated HSPC population (Figure 5). MPPs were the most radiosensitive (Figure 5A,B) and LT-HSPs the most radioresistant (Figure 5E,F); EV-mediated bystander responses were also stronger in the MPP population compared to LT-HSCs. This difference in the cellular response to radiation and EV treatment was reflected in the distribution of the different subpopulations within the HSPC pool. In unirradiated mice, HSPCs were composed of 41% LT-HSCs, 25% ST-HSCs and 34% MPPs. Twenty-four hours after irradiation, a strong dose-dependent redistribution between the LT-HSCs and MPPs could be observed: the fraction of MPPs decreased by 1.1-fold, 2.6-fold and 3.7-fold while the fraction of LT-HSCs increased by 1.1-fold, 1.3-fold and 1.8-fold after 0.1Gy, 0.25Gy and 2Gy irradiation, respectively. The fraction of the ST-HSC subpopulation remained relatively constant (Figure 5G, left bars). Short-term bystander effects induced by EVs isolated 24 h after irradiation led to comparable, albeit milder, effects to direct irradiation: MPPs decreased 1.2-fold, 1.3-fold and 1.5-fold while LT-HSCs increased 1.2-fold, 1.3-fold and 1.6-fold after 0.1Gy, 0.25Gy and 2Gy EV treatment, respectively (Figure 5G, middle bars). Similarly to the unfractionated HSPCs, the studied HSPC subpopulations showed no IR-induced bystander responses if mice were treated with EVs isolated 3 months after irradiation (Figure 5G, right bars). No significant changes were detected in any of the HSPC subpopulations three months after low-dose (0.1Gy and 0.25Gy) irradiation, and a mild, 1.25-fold reduction was seen in the fraction of MPP cells after irradiation with 2Gy, with a respective 1.35-fold increase in the fraction of LT-HSCs (Figure 5H, left bars). Long-term bystander effects (3 months after treatment with EVs isolated 24 h after irradiation) were very similar (1.4-fold decrease of MPPs and 1.3-fold increase in LT-HSCs) to changes seen three months after direct irradiation of mice (Figure 5H, right bars).

#### 3.2.2. Alterations in the LP Pool

Low-dose (0.1Gy and 0.25Gy) irradiation did not alter LP numbers either 24 h or 3 months after irradiation. High-dose (2Gy) irradiation induced a strong, 7.1-fold decrease in LP numbers 24 h after irradiation, and 3 months later LP numbers were still 1.6-fold below control values, indicating a slow and partial regeneration (Figure 6A,B, grey bars). EVs isolated 24 h or 3 months after irradiation could not induce short-term bystander changes in LP numbers (Figure 6A, red and yellow bars). A mild but significant (1.3-fold) decrease in relative LP numbers was measured 3 months after treatment of mice with 2Gy EVs isolated 24 h after irradiation, indicating the development of a long-term bystander response (Figure 6B, blue bars).

#### 3.2.3. Alterations in the GM Pool

Early changes in the GM cell pool were mild, leading to very similar effects in directly irradiated and bystander mice (1.15-fold and 1.2-fold decrease in mice irradiated with 0.25Gy or treated with 0.25Gy EVs isolated 24 h after irradiation, respectively) (Figure 7A grey and red bars). In EV-recipient mice injected with EVs isolated 3 months after irradiation, no bystander effects were found (Figure 7A, yellow bars). Long-term effects were stronger than acute effects. A 1.96-fold decrease in the GM pool was detected in mice 3 months after irradiation with 2Gy. Treatment of mice with 0.25Gy and 2Gy EVs isolated 24 h after irradiation induced a 1.3-fold and 2-fold decrease in the GM pool, respectively (Figure 7B, grey and blue bars).

#### 3.2.4. Alterations in the MSC Pool

MSCs represent important cellular components of non-hematopoietic origin within the BM. Also, it was shown that MSC-derived EVs play an important role in shaping acute radiation damage to the bone marrow [39]. Therefore, we investigated direct radiation and EV-mediated bystander effects on the MSC population, characterized as Lin-Sca1+CD44+ cells [40,41,42]. In directly irradiated mice, the relative number of MSCs decreased 1.9-fold, 1.6-fold and 5-fold after 0.1Gy, 0.25Gy and 2Gy irradiation, respectively, when compared to unirradiated mice 24 h after the total-body exposure (Figure 8A, grey bars). In mice treated with EVs isolated 24 h after irradiation, the MSC pool also decreased, but this decrease was relatively uniform and showed no correlation with the dose used for irradiating the mice from which EVs were isolated. Thus, the relative number of MSCs decreased 1.5-fold in mice treated with 0.1Gy and 0.25Gy EVs and 1.6-fold after 2Gy EVs, compared to 0Gy EV-treated mice (Figure 8A, red bars). Similarly to the other BM subpopulations, MSC numbers did not change in mice treated with EVs isolated from irradiated mice 3 months after exposure (Figure 8A, yellow bars). Apart of a moderate decrease (1.64-fold) in mice irradiated with 2Gy, no further long-term direct IR effects or bystander effects could be measured in the MSC pool (Figure 8B, grey bars).

### 3.3. EVs Can Mediate IR-Induced Apoptosis in the BM in a Bystander Manner

We have shown that IR-induced BM damage could be transferred by EVs in a bystander manner leading to a decrease in the pool of the different hematopoietic stem and progenitor cells. Since apoptosis is a major mechanism of IR-induced cell death in the hematopoietic system, we investigated whether EVs could transmit apoptotic signals to the bystander cells. Mice were either directly irradiated with 0.1Gy, 0.25Gy or 2Gy of IR, or treated with BM-derived EVs isolated from directly irradiated mice and apoptosis frequency was determined in those cell compartments which showed significant short-term bystander responses, namely the HSPC, MSC and LP pool. Since apoptosis is an early event after irradiation, we determined apoptosis frequency 4 h after direct irradiation or EV treatment, based also on our previous findings on the kinetics of apoptosis in hematopoietic and immune cells [43,44]. Apoptosis was seen only following 2Gy irradiation or 2Gy EV treatment. The fraction of apoptotic cells increased 9.7-fold, 7-fold and 2.8-fold in the HSPCs, MSCs and LP cells after irradiation of mice with 2Gy. In bystander mice receiving EVs isolated from the BM of directly irradiated mice, a 5.6-fold and 2.1-fold increase in apoptosis frequency was measured in HSPCs and LP cells, and no apoptosis was seen in the MSCs (Figure 9). These data show that EVs could transmit IR-induced apoptotic signals but only to certain cell types within the BM.

### 3.4. IR and EV Transfer from Irradiated to Bystander Mice Induce Stem Cell Migration from Bone Marrow to the Spleen

Since migration of stem cells into the periphery is a further possible mechanism leading to a decrease in the cell pools within the BM, we analyzed relative changes in the fraction of HSPCs and MSCs in the spleens of mice 24 h after irradiation or EV injection.

A tendency for an increased dose-dependent migration of both HSPCs and MSCs was detected in directly irradiated mice, though changes were statistically only significant for the migration of MSCs in mice irradiated with 2Gy (Figure 10). EV treatment had a moderate effect on BM cell migration into the periphery. With the exception of a significant increase in the splenic fraction of HSPCs in mice treated with 0.25Gy EVs, no further changes were detected in bystander mice (Figure 10).

### 3.5. The miRNA Profile of EVs Isolated 24 Hours and 3 Months after Irradiation Are Different

Previously we performed a miRNA profiling of BM-derived EVs isolated from mice irradiated with low (0.1Gy) and high (2Gy) dose IR [27]. Based on these analyses, we identified 8 miRNAs which were significantly up- or downregulated in the BM-derived EVs 24 h after both low- (0.1Gy) and high (2Gy)-dose irradiation. Since our data show that BM-derived EVs can mediate IR-induced bystander effects only if they are isolated early (24 h) after irradiation and they lose this capacity if they are isolated 3 months after irradiation, we compared expression of these 8 miRNAs in the BM-derived EVs isolated 24 h and 3 months after irradiation by qRT-PCR. All 8 miRNAs showed significantly altered expression profiles, both in 0.1Gy EVs and 2Gy EVs isolated 24 h after irradiation (Figure 11A); the relative concentration of five miRNAs (mmu-miR-33-3p, mmu-miR-200c-5p, mmu-miR-140-3p, mmu-miR-744-3p, mmu-miR-669) decreased while the relative concentration of three miRNAs (mmu-miR-152-3p, mmu-miR-199a-5p, mmu-miR-375-3p) increased after IR exposure. Moreover, changes in the levels of these miRNAs were dose-dependent. To map long-term effects induced by IR in the miRNA expression of BM-derived EVs, the same eight miRNAs were examined in EVs isolated 3 months after the irradiation of the mice. Two miRNAs were significantly decreased in both 0.1Gy and 2Gy EVs (mmu-miR-744-3p and mmu-miR-152-3p), while mmu-miR-375-3p was only decreased in 2Gy EVs. The rest of the miRNAs were not significantly different from 0Gy EVs (Figure 11A). It can be seen that the majority of miRNAs which were altered in EVs isolated 24 h after irradiation were normalized in the EVs isolated 3 months after irradiation. The only miRNA which was downregulated in both 0.1Gy and 2Gy EVs isolated both 24 h and 3 months after irradiation was mmu-miR-744-3p. Importantly, two miRNAs changed in opposite direction in EVs isolated 24 h and 3 months after irradiation. While mmu-miR-152-3p increased significantly and in a dose-dependent manner in EVs isolated 24 h after irradiation, its expression decreased in both 0.1Gy and 2Gy EVs isolated 3 months after irradiation. Mmu-miR-375-3p also increased dose-dependently in EVs isolated 24 h after irradiation, while its expression returned to the control value in 0.1Gy EVs and decreased in 2Gy EVs isolated 3 months after irradiation (Figure 11A). Thus, the expression levels of seven out of the eight miRNAs were different in EVs isolated 3 months after irradiation compared to EVs isolated 24 h after irradiation.

In order to investigate the processes which these miRNAs might regulate in the EV acceptor cells, we performed a pathway analysis using DIANATools miRPath v.3. The seven examined miRNAs altogether could be associated with 24 significantly altered pathways (Appendix A). The majority of these pathways were cancer-related and/or involved in cellular response to IR. Pathways specifically relevant for BM processes (such as signaling pathways for pluripotency of stem cells) were also significantly altered (Figure 11B). Most of these pathways were regulated by more than one miRNA and individual miRNAs were involved in the regulation of multiple pathways (Appendix A).

## 4. Discussion

BM is a particularly radiosensitive tissue, prone to both IR-induced deterministic effects after high-dose exposure (BM insufficiency responsible for the development of hematopoietic syndrome in acute radiation sickness) and IR-induced leukemia developing in a stochastic manner after low doses as well. Intercellular communication is an important mechanism in the proper functioning of the hematopoietic system and alterations in the signaling pathways contribute to the development of BM pathologies. Therefore, BM is a tissue in which IR-induced bystander signals are important in shaping the short- and long-term consequences of BM irradiation. Previously we reported that BM-derived EVs were able to mediate IR-induced bystander effects in the BM. We developed an in vivo model system, where the role of EVs in radiation-induced bystander signals could be studied in the bone marrow. C57Bl/6 mice were irradiated and EVs from the BM supernatant were injected into untreated naïve mice. In this way, we were able to follow radiation-related changes in the hematopoietic system of unirradiated mice transmitted by EVs [25,26,27]. In the present project we intended to characterize EV-mediated bystander effects in more details, namely to determine (a) the most prone cellular subpopulations responding to bystander signals, (b) how long EV-mediated bystander signals persist in EV-treated mice, and (c) how long EVs maintain their capacity to transmit IR-induced bystander signals. In order to do this, early and late direct radiation effects on the BM were compared to EV-mediated effects after treatment of mice with EVs isolated from the BM of irradiated mice.

Growing amounts of evidence support the observation that IR induces EV release both in a dose- and time-dependent manner, due most probably to the activation of additional stress-inducible pathways of EV secretion [45]. Increased EV secretion following irradiation was demonstrated in multiple in vitro studies in aneuploid immortal keratinocytes (HaCaT) [20], human breast adenocarcinoma (MCF7) [18,19], five different glioblastoma cell lines [46] and human prostate cancer cell lines [47]. Li et al. showed that irradiation with 1Gy high linear energy transfer (LET) ions stimulated EV release by about 3-fold in immortalized human bronchial epithelial cells [48]. Arscott et al. observed increased abundance of EVs of human normal astrocytes after exposure to 4Gy [46]. In primary fibroblasts, Elbakrawy et al. found that EV secretion peaked 1 h and 24 h following irradiation [49]. Furthermore, it was demonstrated by Cagatay et al. that various organs (brain, heart, liver) and plasma have an increased rate of EV secretion 1 day and 15 days after total or partial body irradiation with 2Gy [50]. Our results also show a dose-dependent increase in EV production by BM cells 24 h after irradiation of mice. Furthermore, increased particle release could be seen in EVs isolated 3 months after irradiation of mice with a high dose (2Gy), pointing to a persistent deregulation in EV secretion mechanisms.

EV-mediated bystander responses were investigated at two different time points. Early bystander responses were investigated 24 h after EV treatment and the effects were compared to directly irradiated mice 24 h after irradiation. Late bystander responses were investigated 3 months after EV treatment and the effects were compared to effects in mice 3 months after direct irradiation. For both bystander groups, EVs isolated 24 h after irradiation were used. At an early time point (24 h) after irradiation, EV cargo should reflect IR-induced acute molecular damage (such as oxidative stress, DNA damage, or gene expression alterations). A third bystander group was also designed, in which EV treatment was performed with EVs isolated 3 months after irradiation of the mice. The rationale for this group was to see whether at a later phase after irradiation when BM is in the process of regeneration, though residual damage persists secreted EVs can still mediate bystander responses, resembling those seen in directly irradiated mice 3 months after irradiation.

Direct irradiation of mice induced a significant reduction in the pool of most BM cell subpopulations, with the exception of GMs. The LP pool decreased only after high-dose (2Gy) irradiation; however, the HSPC and MSC pool was significantly depleted after low doses as well. Within the HSPC subpopulation, a redistribution between LT-HSCs and MPPs was observed with a dose-dependent decrease in the MPP pool and a correspondent increase in the LT-HSC pool. Three months after irradiation, strong but incomplete regeneration was noted in all cell subpopulations, since moderately decreased cell pools in mice irradiated with 2Gy still persisted. Long-term persistence of reduced stem and progenitor cell pools after moderate-to-high dose irradiation has been reported by other groups as well [51]. The GM pool, which was not affected 24 h after irradiation, was also moderately depleted after 2Gy. These data indicate that radiation damage in stem cells and LPs manifests very rapidly after irradiation, while the effect in GMs is delayed. This is similar to the pattern of radiation-induced cell depletion in the peripheral blood, where radiation induces quick and dose-dependent lymphocyte depletion, while the nadir of cells originating from common myeloid progenitors (granulocytes and platelets) is around one month after irradiation. It is interesting to note the quick and significant depletion in the HSPC pool with no dose-dependency, which was detected by other groups too [52,53]. While high-dose effects are most probably due to radiation-induced increased cell death, as supported by our data showing increased apoptosis, the significant depletion in the HSPC pool after low doses cannot be attributed to increased cell death. Increased migration of stem cells into the periphery [54] might also lead to reduced cell numbers in the bone marrow, but our data do not support this mechanism. Other mechanisms may prevail. Such mechanisms might be increased autophagy in the HSPCs, or a decrease in self renewal. Further studies are needed to reveal the underlying mechanisms.

EV-mediated bystander responses using BM-derived EVs isolated 24 h after irradiation induced comparative cell depletion patterns to direct irradiation. In general, the degree of cell depletion was milder than in the directly irradiated mice, showed no dose-dependency, and mainly manifested in mice treated with 2Gy EVs. Most probably, radiation-induced bystander effects mediated by EVs were realized partly by similar mechanisms to direct irradiation, since EVs could induce similar levels of apoptosis in those cell subpopulations, in which direct irradiation also induced apoptosis. This result is consistent with the findings from Li et al., who proved the ability of serum EVs from high-dose-irradiated mice to induce apoptosis in the gastrointestinal tract of EV-recipient mice [55]. While direct irradiation led to no significant increase in migration of BM stem cells into the spleen, a moderate but significant HSPC migration was demonstrated in mice treated with 0.25Gy EVs. Other studies also reported that EVs of different origins were able to induce cell migration. Cao et al. detected increased stem cell antigen-1 (Sca1) positive cell population in the circulation up to four weeks after irradiation of the mice [56]. There are reports that EVs originating from irradiated tumor cells enhance migration of in vitro cultured non-irradiated tumor cells [46,57].

An important finding was that EV-mediated effects were durable, and 3 months after EV treatment, cell depletion kinetics of most cell subpopulations (with the exception of MSCs and HSPCs) resembled effects seen in directly irradiated mice 3 months after irradiation. Other in vitro studies also reported EV-mediated long-term radiation-induced bystander effects [49]. Thus, we hypothesize that EVs isolated within 24 h of irradiation transmit signals which are able to induce acute cellular damage resembling direct irradiation. While we still lack the knowledge of the mechanisms by which these bystander effects develop, the molecular processes certainly highly resemble and, in part, may even overlap with those induced by direct irradiation. This is supported by the fact that, 3 months after direct irradiation or EV treatment, the recovery status of most of the individual bone marrow subpopulations is very similar, indicating that regeneration kinetics is realized by similar mechanisms. However, in the case of HSPCs, EV-mediated long-term depletion of the cell pool seems to be persistent with no obvious signs of regeneration. Further studies are needed to reveal the long-term molecular processes initiated by EVs in the individual cell populations.

Despite the fact that, 3 months after direct irradiation, mice still exhibited significant residual damage after high-dose irradiation in each studied cell subpopulation, an interesting finding was that EVs isolated 3 months after irradiation could not transmit statistically significant bystander signals manifesting in quantitative changes in the various cell pools. However, an increased inter-individual variability was present in the pool of the different hematopoietic stem and progenitor cells, which was not radiation-dependent but rather EV-dependent. Since EVs used for the treatment of these bystander mice originated from mice 3 months older than all the other bystander groups, it cannot be excluded that aging-related signals were also transmitted by EVs. The fact that EVs are able to transmit aging-related signals was demonstrated by several groups [58,59]. We hypothesize that qualitatively different IR-induced bystander signals are initiated by EVs isolated long-term after irradiation compared to EVs isolated acutely after IR, where mechanisms related to IR-induced premature aging (such as increased senescence, genomic instability, chronic inflammation) are predominant rather than quantitative changes in the various cell pools.

Since both our group [25,27] and others [21] have previously shown that EV-mediated miRNA transfer are involved in IR-induced bystander responses, and that the miRNA content of EVs is influenced by IR [60], we focused on comparing the miRNA content of EVs isolated 24 h and 3 months after irradiation. In our previous study we analyzed the miRNA profile of BM-derived EVs isolated 24 h after irradiation [27]. We identified eight miRNAs which showed dose-dependent changes in their level in the irradiated EVs 24 h after irradiation. Cellular expression of several of these miRNAs was reported to be influenced by IR [61,62]. Since our data show that the level of seven out of the eight miRNAs was normalized in the EVs isolated 3 months after irradiation, we considered these miRNAs important contributors in mediating acute radiation damage in bystander cells. Performing a pathway analysis with the cluster of the seven miRNAs, we found that pathways strongly involved in cellular and molecular response to IR (such as the Hippo, FoxO, PI3K-Akt, Wnt signalling pathways) were significantly altered. As an example, we present our hypothesis on the regulation of the Wnt signalling pathway targeted by five miRNAs differentially expressed in the BM-derived EVs isolated 24 h after irradiation, where the expression pattern of the five miRNAs indicates suppression of the Wnt pathway (Figure 12). Further validation of the effects of the differentially expressed miRNAs on their target proteins of the Wnt pathway are needed to confirm our hypothesis.

This study did not perform a detailed screening of the content of EVs isolated 3 months after irradiation, though it might be interesting to investigate whether EVs emitted by BMCs harboring radiation-induced residual DNA damage can transmit signals involved in the initiation of a malignant process (such as signals leading to genomic instability or triggering inflammatory reactions in the BM microenvironment).

## 5. Conclusions

In conclusion, in this study we showed that BM-derived EVs isolated from irradiated mice could transmit radiation-induced acute damage in the BM of bystander mice, depleting the same cell subpopulations as direct irradiation. We also demonstrated that EV-induced BM alterations were persistent and that, 3 months after EV injection, BM damage was similar to direct radiation damage seen in mice 3 months after irradiation, suggesting similar regeneration kinetics. An important finding was that for initiating bystander signals BM-derived EVs had to be isolated soon after irradiation, since EVs isolated 3 months after irradiation lost their capacity to transmit radiation damage to bystander cells. However, further investigations are needed to determine the time frame within which EVs retain their capacity to mediate acute radiation effects. We identified seven miRNAs which might have important roles in the EV-mediated effects in the BM, since their amount was modified in a dose-dependent manner only in EVs isolated 24 h after irradiation and pathway analysis indicated their contribution in regulating key processes involved in cellular and molecular response to IR. Our data highlight the importance of intercellular communication in the development and modulation of acute radiation damage in the BM.

## Figures and Tables

**Figure 1 cells-11-00155-f001:**
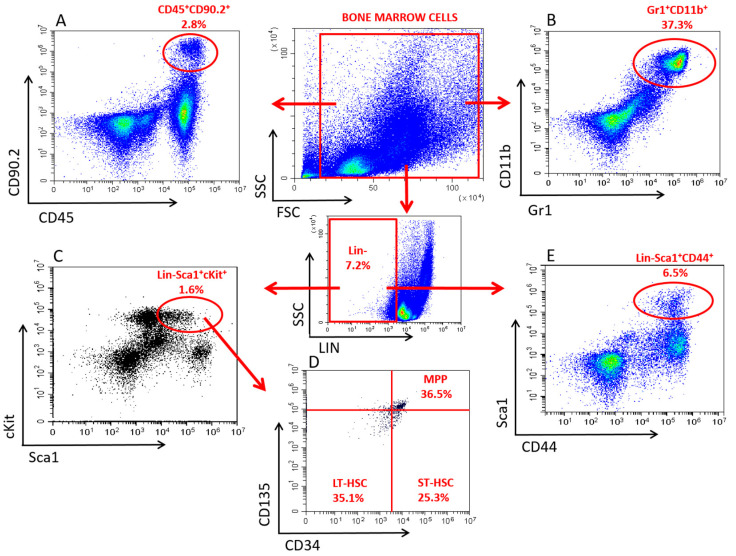
Gating Strategy for the Identification of Different Bone Marrow Cell Populations by Flow Cytometry. (**A**) Lymphoid progenitor cells were identified as CD45 and CD90.2 double positive cells in bone marrow cells. (**B**) Granulocytes/monocytes were characterized as Gr1 and CD11b double positive cells in bone marrow cells. (**C**) Hematopoietic stem and progenitor cells were identified as Sca1 and cKit double positive cells in the Lineage negative bone marrow cells. (**D**) Hematopoietic stem cell subpopulation characterization was done by using the CD34 and CD135 markers in the hematopoietic stem and progenitor cell pool. Long-term hematopoietic stem cells were identified as CD34 and CD135 double negative cells, short-term hematopoietic stem cells were CD34+CD135- cells, multipotent progenitors were characterized as CD34 and CD135 double positive cells. (**E**) Mesenchymal stem and stromal cells were identified as Sca1 and CD44 double positive cells in the lineage-negative bone marrow cells.

**Figure 2 cells-11-00155-f002:**
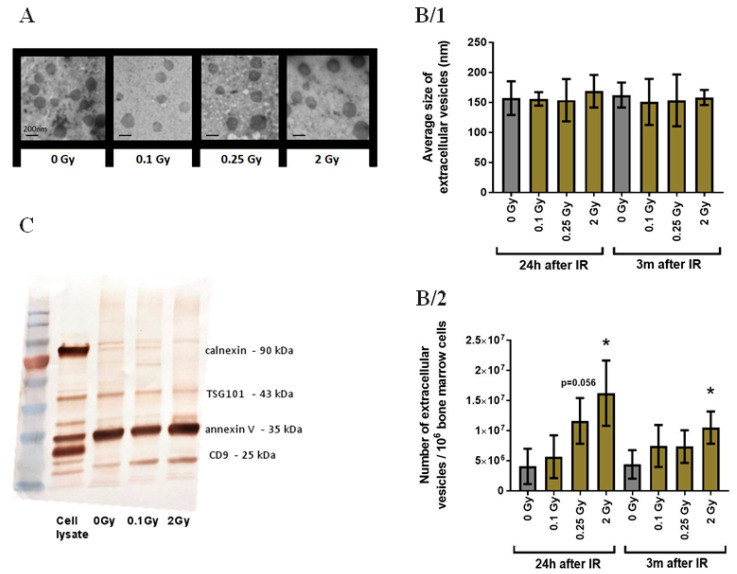
Characterization of Bone Marrow-Derived Extracellular Vesicles. (**A**) Representative transmission electron microscopy images of extracellular vesicles isolated from the bone marrow of mice irradiated with the indicated doses of ionizing radiation. (**B**) Size and concentration of extracellular vesicle suspensions were examined by tunable resistance pulse sensing. Mean values of extracellular vesicle size (**B/1**) and mean extracellular vesicle particle numbers released by 10^6^ bone marrow cells (**B/2**) are shown with bars representing standard deviations (SD). *n* = 3, significance tested by Student’s *t*-test, *p* * < 0.05. (**C**) Representative Western blot analysis of whole cell lysates and extracellular vesicles isolated from the bone marrow of mice irradiated with the indicated doses of ionizing radiation. Lane 1: protein ladder, lane 2: whole cell lysate, lane 3: extracellular vesicle sample from unirradiated mice, lane 4-5: extracellular vesicle samples from mice irradiated with 0.1Gy and 2Gy.

**Figure 3 cells-11-00155-f003:**
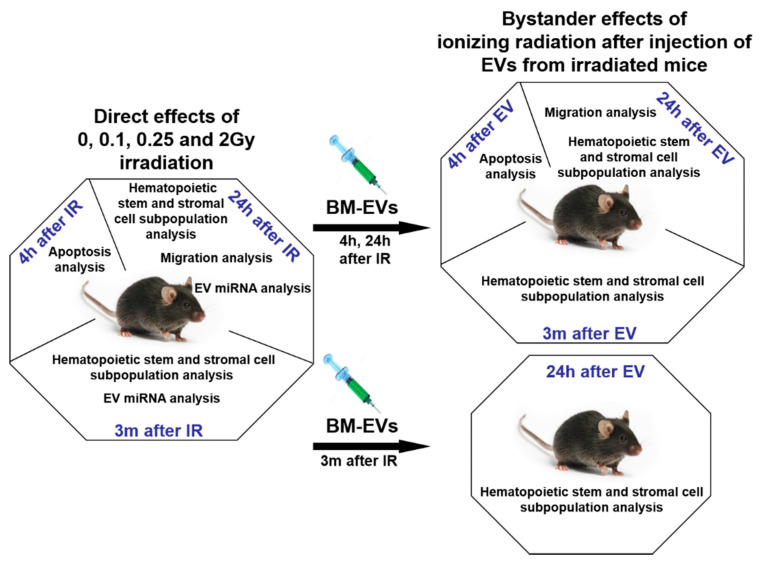
Schematic Presentation of the Workflow of the Study. C57Bl/6 mice were total-body irradiated with different doses (0Gy, 0.1Gy, 0.25Gy and 2Gy) of ionizing radiation. Mice were humanely killed 4 h, 24 h or 3 months later and the bone marrow and spleen were collected. Bone marrow-derived extracellular vesicles were isolated from the bone marrow supernatant of age-matched control and irradiated mice. Bystander effects were monitored in non-irradiated, healthy mice after injecting them with bone marrow-derived extracellular vesicles. Bystander mice were humanely killed 4 h, 24 h or 3 months after extracellular vesicle injection and the same organs were harvested as from the directly irradiated animals. Apoptosis in the bone marrow stem and progenitor cells was measured by TUNEL assay. Bone marrow hematopoietic stem, progenitor and stromal cell subpopulations were characterized phenotypically by flow cytometry. MiRNA expression of BM-EVs was investigated by qRT-PCR.

**Figure 4 cells-11-00155-f004:**
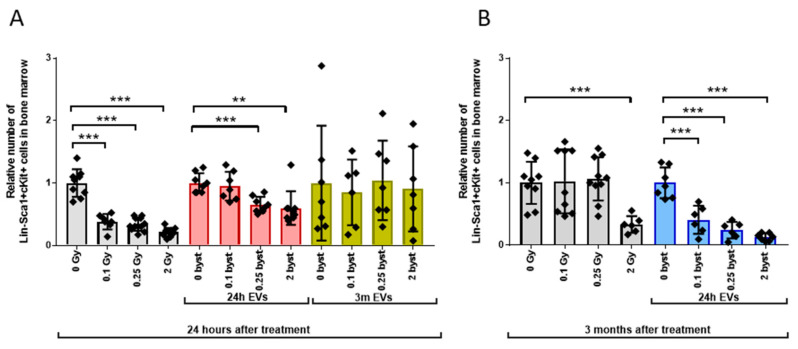
Bone Marrow-Derived Extracellular Vesicles from Irradiated Mice Induce Long-Term Hematopoietic Stem Cell Depletion in the Bone Marrow of Bystander Mice. Irradiation, extracellular vesicle treatment and bone marrow cell phenotyping were carried out as described in the Materials and Methods section. Lin-Sca1+cKit+ cells were considered hematopoietic stem cells. (**A**) Relative changes in hematopoietic stem cell numbers were evaluated 24 h after treatment. Grey bars represent total-body irradiated mice; red bars represent mice treated with bone marrow-derived extracellular vesicles isolated from mice 24 h after irradiation; yellow bars represent mice treated with bone marrow-derived extracellular vesicles isolated from mice 3 months after irradiation. (**B**) Relative changes in hematopoietic stem cell numbers were evaluated 3 months after treatment. Grey bars represent total-body irradiated mice; blue bars represent mice treated with bone marrow-derived extracellular vesicles isolated from mice 24 h after irradiation. Bars represent mean values of relative hematopoietic stem cell numbers, dots show individual values, error bars represent SD. *n* = 6–11, significance tested by Student’s *t*-test, *p* ** < 0.01, *p* *** < 0.001.

**Figure 5 cells-11-00155-f005:**
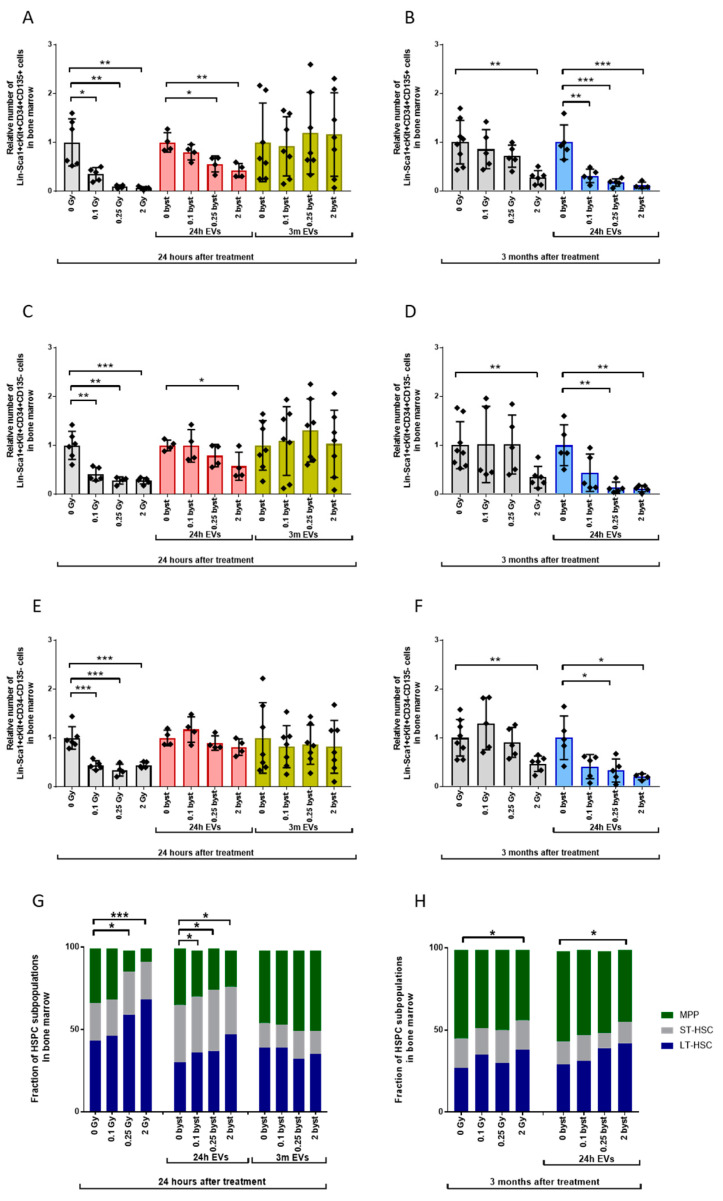
Both Direct Irradiation and Extracellular Vesicles-Mediated Bystander Effects Induce Persistent Redistribution between Multipotent Progenitors and Long-Term Hematopoietic Stem Cells in the Bone Marrow. Irradiation, extracellular vesicle treatment and bone marrow cell phenotyping was carried out as described in the Materials and Methods section. Cellular subpopulations were identified as Lin-Sca1+cKit+CD34-CD135- long-term hematopoietic stem cells (LT-HSC), Lin-Sca1+cKit+CD34+CD135- short-term hematopoietic stem cells (ST-HSC) and Lin-Sca1+cKit+CD34+CD135+ multipotent progenitor cells (MPP) [36,37] and were measured by flow cytometry. (**A**) multipotent progenitors; (**C**) short-term hematopoietic stem cells; (**E**) long-term hematopoietic stem cells. Relative changes in hematopoietic stem cell subpopulations were evaluated 24 h after treatment. Grey bars represent total-body irradiated mice; red bars represent mice treated with bone marrow-derived extracellular vesicles isolated from mice 24 h after irradiation; yellow bars represent mice treated with bone marrow-derived extracellular vesicles isolated from mice 3 months after irradiation. (**B**) multipotent progenitors; (**D**) short-term hematopoietic stem cells; (**F**) long-term hematopoietic stem cells. Relative changes in hematopoietic stem cell numbers were evaluated 3 months after treatment. Grey bars represent total-body irradiated mice; blue bars represent mice treated with bone marrow-derived extracellular vesicles isolated from mice 24 h after irradiation. (**G**) Distribution of individual subpopulations within the hematopoietic stem and progenitor cells was evaluated 24 h after treatment. Left columns represent total-body irradiated mice; middle columns represent mice treated with bone marrow-derived extracellular vesicles isolated from mice 24 h after irradiation; right columns represent mice treated with bone marrow-derived extracellular vesicles isolated from mice 3 months after irradiation. (**H**) Distribution of individual subpopulations within the hematopoietic stem and progenitor cells was evaluated 3 months after treatment. Left columns represent total-body irradiated mice; right columns represent mice treated with bone marrow-derived extracellular vesicles, which were isolated from mice 24 h after irradiation. Bars represent mean fraction of subpopulations, dots show individual values, error bars represent SD. *n* = 6–8, significance tested for individual subtypes of hematopoietic stem cells by Student’s *t*-test, *p* * < 0.05, *p* ** < 0.01, *p* *** < 0.001.

**Figure 6 cells-11-00155-f006:**
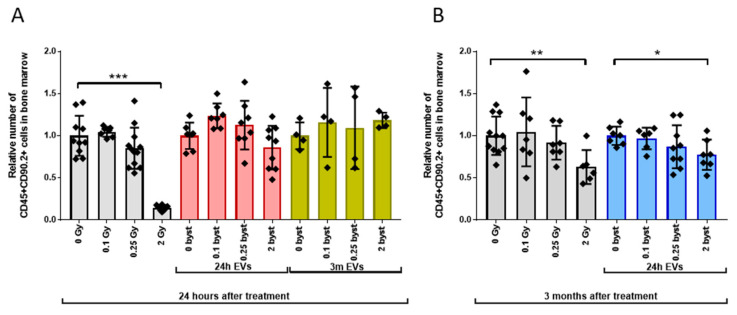
High Dose Irradiation Induces Persistent Depression of Lymphoid Progenitor Numbers, Extracellular Vesicle-Mediated Mild Bystander Effects Develop Late after Treatment. Irradiation, extracellular vesicle treatment and bone marrow cell phenotyping was carried out as described in the Materials and methods section. CD45+CD90.2+ cells were considered lymphoid progenitors. (**A**) Relative changes in lymphoid progenitor numbers were evaluated 24 h after treatment. Grey bars represent total-body irradiated mice; red bars represent mice treated with bone marrow-derived extracellular vesicles isolated from mice 24 h after irradiation; yellow bars represent mice treated with bone marrow-derived extracellular vesicles isolated from mice 3 months after irradiation. (**B**) Relative changes in lymphoid progenitor numbers were evaluated 3 months after treatment. Grey bars represent total-body irradiated mice; blue bars represent mice treated with bone marrow-derived extracellular vesicles isolated from mice 24 h after irradiation Bars represent mean values of relative lymphoid progenitor numbers, dots show individual values, error bars represent SD. *n* = 5–11, significance tested by Student’s *t*-test, *p* * < 0.05, *p* ** < 0.01, *p* *** < 0.001.

**Figure 7 cells-11-00155-f007:**
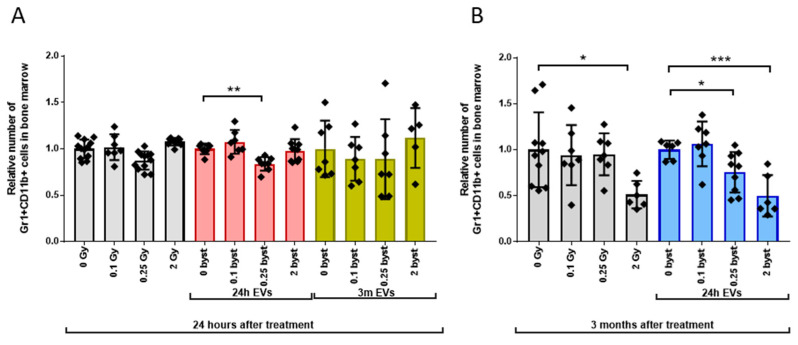
Long-Term Direct Radiation and Extracellular Vesicles-Mediated Bystander Effects on the Granulocytes/Monocytes Pool Are More Severe than Acute Effects. Irradiation, extracellular vesicles treatment and bone marrow cell phenotyping was carried out as described in the Materials and methods section. Gr1+CD11b+ cells were considered granulocytes/monocytes. (**A**) Relative changes in the numbers of granulocytes/monocytes were evaluated 24 h after treatment. Grey bars represent total-body irradiated mice; red bars represent mice treated with bone marrow-derived extracellular vesicles isolated from mice 24 h after irradiation; yellow bars represent mice treated with bone marrow-derived extracellular vesicles isolated from mice 3 months after irradiation. (**B**) Relative changes in the numbers of granulocytes/monocytes were evaluated 3 months after treatment. Grey bars represent total-body irradiated mice; blue bars represent mice treated with bone marrow-derived extracellular vesicles isolated from mice 24 h after irradiation. Bars represent mean values of relative granulocytes/monocytes progenitor numbers, dots show individual values, error bars represent SD. *n* = 5–11, significance tested by Student’s *t*-test, *p* * < 0.05, *p* ** < 0.01, *p* *** < 0.001.

**Figure 8 cells-11-00155-f008:**
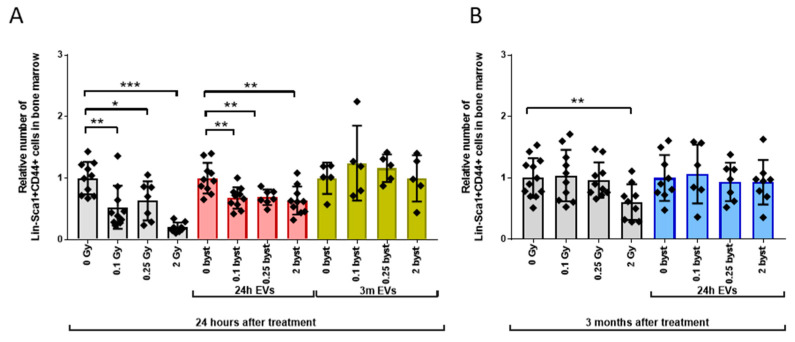
Short-Term Direct Irradiation and Extracellular Vesicle-Mediated Bystander Effects Affecting the Mesenchymal Stromal Cell Pool Are More Severe than Long-Term Effects. Irradiation, extracellular vesicle treatment and bone marrow cell phenotyping was done as described in the Materials and Methods section. Lin-Sca1+CD44+ cells were considered mesenchymal stromal cells. (**A**) Relative changes in mesenchymal stromal cell numbers were evaluated 24 h after treatment. Grey bars represent total-body irradiated mice; red bars represent mice treated with bone marrow-derived extracellular vesicles isolated from mice 24 h after irradiation; yellow bars represent mice treated with bone marrow-derived extracellular vesicles isolated from mice 3 months after irradiation. (**B**) Relative changes in mesenchymal stromal cell numbers were evaluated 3 months after treatment. Grey bars represent total-body irradiated mice; blue bars represent mice treated with bone marrow-derived extracellular vesicles isolated from mice 24 h after irradiation. Bars represent mean values of relative mesenchymal stromal cell numbers, dots show individual values, error bars represent SD. *n* = 5–11, significance tested by Student’s *t*-test, *p* * < 0.05, *p* ** < 0.01, *p* *** < 0.001.

**Figure 9 cells-11-00155-f009:**
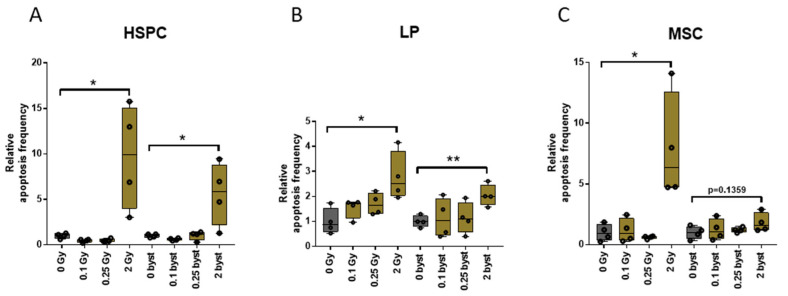
Transfer of Bone Marrow-Derived Extracellular Vesicles from Irradiated Mice Is Able to Induce Apoptosis in a Bystander Manner in Hematopoietic Stem Cells and Lymphoid Progenitors but Not Mesenchymal Stromal Cells. Bone-marrow single-cell suspension was prepared from directly irradiated and extracellular vesicle-treated mice 4 h after treatment and apoptosis was measured by the Tunnel assay as described in the Materials and Methods section. The relative change in apoptosis frequency compared to control mice (either non-irradiated or treated with extracellular vesicles originating from the bone marrow of non-irradiated mice) is shown for hematopoietic stem and progenitor cells (**A**), lymphoid progenitors (**B**) and mesenchymal stromal cells (**C**). Mean, minimum and maximum values are shown, error bars represent SD. *n* = 4, significance tested by Student’s *t*-test, *p* * < 0.05; *p* ** < 0.01.

**Figure 10 cells-11-00155-f010:**
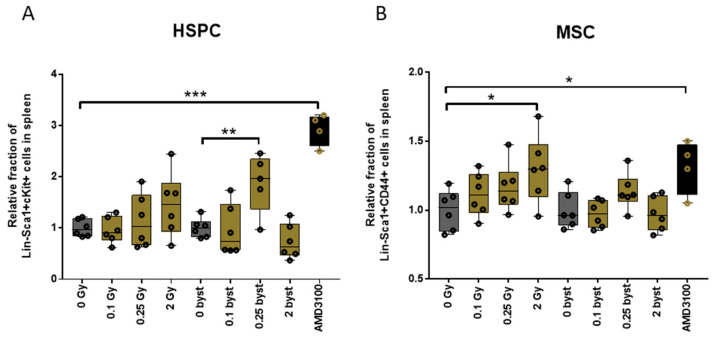
Bone Marrow-Derived Extracellular Vesicles from Low-Dose Irradiated Mice Induce Migration of Hematopoietic Stem and Progenitor Cells into the Spleen in Bystander Mice. Lin-Sca1+cKit+ hematopoietic stem and progenitor cells (**A**) and Lin-Sca1+CD44+ mesenchymal stromal cells (**B**) in the spleen were measured by flow cytometry 24 h after irradiation or injection of extracellular vesicles. Isolation of spleens and splenocyte phenotyping was performed as described in the Materials and Methods section. Black bars represent positive control mice treated with AMD3100. Mean, minimum and maximum values are shown, error bars represent SD. *N* = 6, significance tested by Student’s *t*-test, *p* * < 0.05, *p* ** < 0.01, *p* *** < 0.001.

**Figure 11 cells-11-00155-f011:**
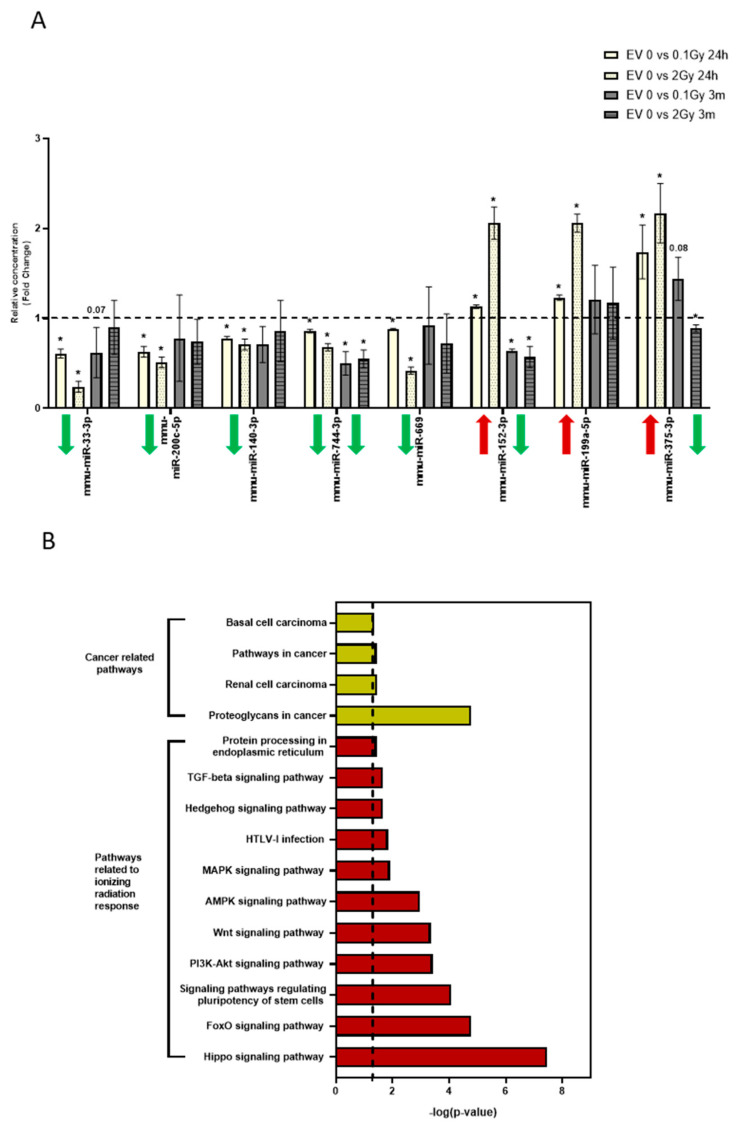
Pathways Related to Cellular Response to Ionizing Radiation Prevail in miRNAs Differentially Expressed in the Bone Marrow-Derived Extracellular Vesicles Isolated from Mice 24 Hours after Irradiation. (**A**) Extracellular vesicles were isolated from the bone marrow of mice, miRNAs purified from extracellular vesicles and the relative concentration of miRNAs was measured by qRT-PCR as described in the Materials and Methods section. *n* = 3; * indicate significant changes (*p* ˂ 0.05) compared to control (0Gy extracellular vesicles samples). Arrows show increased (red arrows) or decreased (green arrows) expression in miRNAs from extracellular vesicles isolated 24 h or 3 months after irradiation. (**B**) KEGG analysis of differentially expressed 7 miRNAs in murine bone marrow-derived extracellular vesicles isolated 24 h but not 3 months after irradiation. A pathway was considered significant if the *p*-value was ˂0.05 (−log10 (0.05) indicated by the dashed line.

**Figure 12 cells-11-00155-f012:**
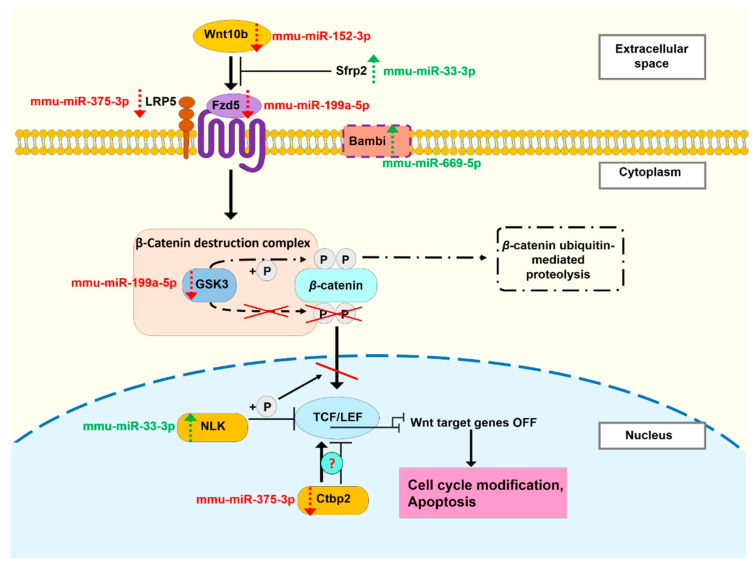
The Influence of Differentially Expressed miRNAs on the Wnt Pathway. The Wnt signal transduction pathway is an evolutionary conserved pathway regulating basic developmental processes such as progenitor cell proliferation and cell-fate specification [63], and it can be downregulated upon exposure to ionizing radiation [64,65]. The transcription of Wnt-related genes can be regulated by the cytoplasmic concentration of the β-catenin intracellular signal transducer. Without Wnt ligand binding to its receptor Frizzeld [66] and co-receptor low-density lipoprotein receptor related 5/6 (LRP5/6) [67,68,69], β-catenin is degraded by a destruction complex, which contains axin, adenomatosus polyposis coli (APC), protein phosphatase 2A (PP2A), and two kinases, glycogen synthase kinase 3 (GSK3) and casein kinase 1α (CK1α). In the β-catenin destruction complex, GSK3 and CK1α phosphorylate β-catenin, leading to its proteosomal degradation [70]. Upon Wnt ligand binding, the destruction complex is disrupted, and β-catenin is enriched in the cytoplasm, which leads to its nuclear transport. In the nucleus β-catenin binds to lymphoid enhancer-binding factor/T-cell factor (LEF/TCF) proteins, transforming it into a transcriptional activator, leading to the transcription of Wnt target genes [71,72,73,74]. In the absence of β-catenin, LEF/TCF block the transcription of Wnt target genes [75,76]. The Wnt pathway was targeted by multiple differentially expressed miRNAs in the extracellular vesicles. Targets were associated with miRNAs by Diana mirPath v.3. We present our hypothesis of how differentially expressed miRNAs in 2Gy extracellular vesicles isolated from the bone marrow of mice 24 h after irradiation lead to the repression of the Wnt pathway. MiRNAs mostly repress the expression of their targets, so Wnt components targeted by upregulated miRNAs (illustrated in red in the Figure) are supposed to be downregulated, and components targeted by downregulated miRNAs (illustrated in green in the Figure) are supposed to be upregulated, compared to 0Gy samples. Arrows indicate the possible changes in the expression of the proteins targeted by the differentially expressed miRNAs: a red down arrow indicates decreased expression upon miRNA interaction, and a green up arrow indicates increased target expression. Mmu-miR-33-3p, a down-regulated miRNA targets a secreted frizzled-related protein 2 (Sfrp2), which is a Wnt antagonist [77,78], and nemo-like kinase (NLK), which is an inactivator of β-catenin TCF/LEF transcription complex formation [79]. The other downregulated miRNA mmu-miR-669o-5p interacts with the BMP and Activin Membrane Bound Inhibitor (Bambi) which can both up- and downregulate the β-catenin signaling [80,81,82]. The upregulated mmu-miR-152-3p targets Wnt10b, which promotes the β-catenin-dependent Wnt signaling pathway [83]. Wnt receptors are also targeted by upregulated miRNAs: LRP5 by mm-miR-375-3p and FZD5 by mmu-miR-199a-5p. Two Wnt inhibitors are targeted by two upregulated miRNAs: GSK-3β by mmu-miR-199a-5p, and Ctbp2 by mmu-miR-375-3p. Ctbp2 is originally an inhibitor, but some studies indicate that it may also act as an activator of TCF [84,85]. Thus, it can be seen that downregulated miRNAs target inhibitors, while upregulated miRNAs mostly target Wnt pathway initiators, which indicate an overall downregulation of the Wnt signaling pathway.

## Data Availability

Not applicable.

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
