# Peer review of "Extracellular Vesicles Derived from Bone Marrow in an Early Stage of Ionizing Radiation Damage Are Able to Induce Bystander Responses in the Bone Marrow"

_cells, 2022, doi:10.3390/cells11010155_

Round 1

Reviewer 1 Report

Thank you very much for this interesting paper. I only have a few minor comments and questions: 

In the statistics section you have mentioned that you performed students t test for statistical analysis. Would you please be able to include which tets was used to test for normal distribution etc? 

I liked how you included single values in Figure 3, 5, 6, and 7. Would it be possible to do the same in the other figures? 

In general for all the figures: I am unsure if it is the review quality or if it is  a general problem, but a lot of the graphs (especially axis labels) and e.g. Figure 11 seem very blurry, especially when printed. Could you  please try to improve  quality of these figures? 

Reviewer 2 Report

The manuscript entitled “Extracellular vesicles derived from bone marrow in an early stage of ionizing radiation damage are able to induce bystander responses in the bone marrow” is focusing on the ability of extracellular vesicles (EV), isolated from irradiated bone marrow, to transfer bystander effects to hematopoietic and stromal cells from non-irradiated mice. Three doses of X-rays ranging from 0.1 Gy to 2 Gy have been used and EV have been extracted either 24h or 3 months after irradiation.

This manuscript demonstrates that:

  • Only EV that are isolated early time post-irradiation are able to transmit bystander effect;
  • HSPC pool is reduced long-term after injection of irradiated EV whatever the dose;
  • Lymphoid progenitor pool is altered long-term after injection of irradiated EV only at 2 Gy;
  • Myeloid cell pool is reduced long-term after injection of irradiated EV at 0.25 Gy and 2 Gy;
  • A MSC subpopulation is diminished only short-term after injection of irradiated EV.

Finally, authors describe a different miRNA profile in EV isolated either 24h or 3 months after irradiation and hypothesize that the bystander effects due to EV (isolated early after irradiation) could be due to a down-regulation of the Wnt signaling pathway.

Comments to Authors

This manuscript demonstrates that secretion of EV in bone marrow is increased 24h after irradiation in a dose-dependent manner and that this increase persists, but at a lower extend, 3 months after irradiation. Comparative studies of Total body irradiated mice versus non-irradiated mice injected with EV show that only early isolated EV are able to induce a bystander effect on bone marrow HSPC, lymphoid progenitors, myeloid cells and MSC by decreasing their number.

In order to explain these results, apoptosis and migration of HSPC and MSC into the spleen have been studied. Finally, to understand why only EV isolated early time after irradiation induce a bystander effect, a comparative study of the 24h- and 3 month-EV miRNA profiles was performed.

This manuscript is clear and well-written, but I think that some data would be strengthened by at least a few easy experiments. There are also some mistakes/inaccuracies to correct in the manuscript. Moreover, I think that some illustrations of FACS plots are necessary for a better understanding.

Abstract and Introduction are clear and well-documented.

Methods. Description of different materials and methods is sufficiently precise and clear.

Results

Results are presented globally in a clear and precise manner. However, I think that some data are missing and others must be deeply explained.

Paragraph 3.2.1:

Lin-Sca1+cKit+ cells are usually considered as HSPC (Hematopoietic stem and progenitor cells) and not as HSC. MPP (multipotent progenitors) are included in this population of cells. I would replace ‘HSC” by “HSPC” in the text.

In Figure 3 left panel, measure of KLS number after 3m EVs shows a great variability indicating 2 different responses: a group of mice with a bystander effect and a group of mice without a bystander effect.  Did the authors compare the bone marrow phenotype from the group showing a reduced number of KLS and those showing a normal (or higher) number of KLS? How could you explain this phenomenon?

Line 304: 3 months after treatment, you mentioned an interesting stronger decrease of the KLS number in mice injected with low-dose irradiated EV compared to directly irradiated mice. That means that bystander effect due to EV after low doses is more deleterious on KLS than a low-dose total body irradiation. Authors do not speculate/discuss this observation.

It would be interesting to analyze several other radio-induced parameters in KLS by comparing direct effects of irradiation and EV bystander effects. Did the authors measure intracellular ROS, gH2AX foci (kinetics) or p53 target genes (qRT-PCR) in KLS from the different groups?

Figure 4: A representative FACS plot is necessary to visualize the 3 different populations of HSPC that you analyzed.

Authors have to be careful about the HSPC phenotyping. Cells that authors call ST-HSC (CD135- CD34+) in the manuscript are, in fact, a mix of ST-HSC, MPP2 and MPP3 (Pietras, 2015; PMID: 26095048), which can be discriminated with both CD48 and CD150 markers.

Representation of these data in proportions are not sufficiently informative as the bone marrow cellularity is missing. This graphical representation does not include the fact that HSC number is very significantly decreased 24h after a 2Gy-TBI. I would show the absolute number of each sub-population.

 Paragraph 3.2.2: A representative FACS plot would be welcome.

 Paragraph 3.2.3:

A phenotype based on GR1+ and CD11b+ criteria is absolutely not representative of GMP in bone marrow. GR1+ CD11b+ cells in bone marrow are mature myeloid cells. You can discriminate GMP with the following criteria: Lin- Sca1- cKit+ CD34+ CD16/32+. Authors must replace “GMP” by “mature myeloid cells”.

 Paragraph 3.2.4:

Why did the authors choose to look at a subpopulation of MSC in this context of bystander effect?

The chosen criteria Lin- Sca1+ CD44+ is representative of a large population of MSC. To better discriminate bone marrow MSC, authors should add at least the CD45- CD31- criteria.

 Paragraph 3.3:

Why did the authors analyze apoptosis on KLS, which is a mix of LT-HSC, ST-HSC and progenitors. Which HSPC subpopulation is more apoptotic after 2 Gy? MPP?

Why did the authors choose the timing of 4h post-treatment?

 Paragraph 3.4:

Why did the authors not look at the HSC mobilization to the peripheral blood? Why did they prefer to look at the spleen?

Discussion

Some results are well discussed, but some others could be even more.

Line 693, authors discuss about the strong depletion in HSPC population after low doses, which is not due to apoptosis and not due to migration to the spleen. Authors hypothesized a migration to peripheral blood, but did not look at it. These data are easy to obtain and should be described (at least 24h post-treatment) in this manuscript, as there is no other hypothesis about this HSPC decrease.

Line 745, as an example, authors hypothesized that the bystander effect due to early isolated EV pass through a down-regulation of the Wnt pathway. It is known that inactivation of the Wnt signalization leads to a proteasomal or autophagic degradation. What do the authors think about this possibility to explain the decreased number of HSPC?
